



# Dispersal of bacteria and stimulation of permafrost decomposition by Collembola

Sylvain Monteux[1,2], Janine Mariën[3], Eveline J. Krab[1]

[1]Department of Soil and Environment, Sveriges Lantbruksuniversitet SLU, Uppsala, 750 07, Sweden
[2]Department of Environmental Science, Stockholms Universitet, Stockholm, 106 91, Sweden
[3]Department of Animal Ecology, Vrije Universiteit Amsterdam, Amsterdam, 1081 HV, The Netherlands

*Correspondence to*: Sylvain Monteux (monteux@pm.me)

**Abstract.** Contrary to most soils, permafrost soils have the atypical feature of being almost entirely deprived of soil fauna. Abiotic constraints on the fate of permafrost carbon after thawing are increasingly understood, but biotic constraints remain
scarcely investigated. Incubation studies, essential to estimate effects of permafrost thaw on carbon cycling, typically measure the consequences of permafrost thaw in isolation from the topsoil, and thus do not account for the effects of altered biotic interactions because of e.g. colonization by soil fauna. Microarthropods facilitate the dispersal of microorganisms in soil, both on their cuticle (ectozoochory) and through their digestive tract (endozoochory), which may be particularly important in permafrost soils, considering that microbial community composition can strongly constrain permafrost
biogeochemical processes.

Here we tested how a model species of microarthropod (the Collembola *Folsomia candida*) affected aerobic $CO_2$ production of permafrost soil over a 25 days incubation. By using collembola stock cultures grown on permafrost soil or on an arctic topsoil, we aimed to assess the potential for endo- and ectozoochory of soil bacteria, while cultures grown on gypsum and sprayed with soil suspensions would allow to observe only ectozoochory.

The presence of collembola or the different treatments imposed to the collembola microbiome (growth substrate, spraying) did not alter bacterial community composition as a whole (relative abundances, weighted UniFrac). However, when Collembola were present (independent of their treatment), a number of introduced bacteria were found (presence-absence, unweighted UniFrac), resulting in increased species richness.

$CO_2$ production was increased by 25.85% in the presence of collembola, about half of which could be attributed to
collembola respiration. We argue that the remaining 13.22% (95% CI:3.2-23.25%) can be considered a priming effect of the presence of collembola, i.e. a stimulation of permafrost $CO_2$ production in the presence of active microarthropod decomposers. Overall, our findings underline the importance of biotic interactions in permafrost biogeochemical processes, and the need to explore the additive or interactive effects of other soil food web groups of which permafrost soils are deprived.



## 1. Introduction

Carbon fluxes from soils are largely governed by the rate of decomposition of organic matter. Soil fauna are a crucial component in this decomposition (García-Palacios et al., 2013; Griffiths et al., 2021), both directly through mechanical degradation of litter and corpses into smaller pieces, and indirectly through feeding habits controlling the abundances of other decomposer groups such as fungi, microbial eukaryotes or bacteria (Hanlon and Anderson, 1979; Kaneda and Kaneko, 2008; Frouz et al., 2020; Potapov et al., 2020). In Arctic soils, microarthropods and in particular Collembola are particularly impactful through this latter mechanism.

Contrary to other important groups of soil fauna, such as earthworms, millipedes (Golovatch and Kime, 2009; Berman et al., 2015) or woodlice (Sfenthourakis and Hornung, 2018), collembola are ubiquitous in Arctic soils where they can reach high densities – up to 130,000 individuals per m$^2$ in high arctic Greenland (Sørensen et al., 2006). However, they are mostly abundant in the topsoil, and to our best knowledge have never been observed in the perennially-frozen subsoil, the permafrost. Permafrost soils are a prominent feature of Arctic landscapes, often considered from the perspective of the huge carbon stock they represent and the positive feedback to climate change that their thawing will likely induce (IPCC, 2021). The frozen conditions over long periods of time have eliminated most fauna and non-microbial life from these environments, and despite the possibility of resuscitating such organisms such as plants (Yashina et al., 2012), nematodes (Shatilovich et al., 2018) or rotifers (Shmakova et al., 2021), newly-thawed permafrost soils generally harbour an extremely simplified soil food web entirely deprived of metazoans. Despite this particularity, in the very active field of permafrost research the absence of collembola and the consequence of their possible introduction on biogeochemical cycling has been mostly overlooked.

In contrast to collembola, some microorganisms survive and/or thrive in permafrost conditions, and microbial adaptations to frozen conditions have been studied (Mackelprang et al., 2011, 2017; Hultman et al., 2015; Bottos et al., 2018). However, not all microbes survive these conditions, and the combination of environmental constraints exerted over long periods of time (Mackelprang et al., 2017) and strong dispersal limitations (Bottos et al., 2018) result in microbial communities that can be deprived of some functions (Knoblauch et al., 2018; Monteux et al., 2020; Barbato et al., 2022). The re-introduction of such functions can result in drastic changes in permafrost processes, and sizable impacts on greenhouse gas production have been observed in vitro for $CH_4$ and $CO_2$ (Knoblauch et al., 2018; Monteux et al., 2020) and confirmed in situ for $N_2O$ (Marushchak et al., 2021). Upon thawing, this re-introduction of missing function, or ecological rescue (Calderón et al., 2017) requires microorganisms to migrate into this newly-available habitat, which could happen for instance laterally through airborne dispersal (Harding et al., 2011) – e.g. for permafrost exposed to the air in abrupt thaw processes (Inglese et al., 2017) – or vertically through percolation in the soil column when the active layer becomes deeper (Monteux et al., 2018; Johnston et al., 2019).





Collembola are another possible vector for the dispersal of microorganisms into newly-thawed permafrost (Buse et al.,
2014). In active layer deepening, collembola migration into newly-thawed permafrost is unlikely since they mostly reside in
topsoil layers, but it will likely occur in soil mixing events (Väisänen et al., 2020) such as thermokarst, active layer
detachment or thaw slump processes where newly-thawed permafrost is exposed to surface conditions. Like all large
organisms, collembola host a variety of microorganisms, their microbiome (Agamennone et al., 2015; Leo et al., 2021).
Because micro-arthropods, such as collembola, can move across large distances compared to fungi or bacteria, the microbial
species in collembolan microbiomes might be among the first to colonize and establish into newly-thawed permafrost when
collembola access it. This could occur through a combination of two main processes: ectozoochory, where the microbiome
of the cuticle disperses into the new habitat, and endozoochory where microorganisms disperse after transiting through the
gut of the animals. To our best knowledge, this has not been explored yet.

We incubated permafrost from the Yedoma domain, which represents a large carbon stock (Strauss et al., 2017) and has
previously been shown to lack certain microbial functions (Monteux et al., 2020), in the presence or absence of a model
species of microarthropod (collembola *Folsomia candida*), and assessed $CO_2$ production and bacterial community
composition over a 25 days aerobic incubation. We used *Folsomia candida* collembola from a stock culture, as well as
collembola subjected to manipulation of their cuticle microbiome, or cuticle and gut microbiomes, to test the following
hypotheses:

1. Collembola presence will alter bacterial community composition, through their grazing.
2. Collembola more closely exposed to topsoil bacteria will change permafrost bacterial community composition
   further than those exposed less intricately. In other words, we expect a gradual change between clean collembola,
   collembola with cuticle microbiome manipulated (ectozoochory) and collembola with both gut and cuticle
microbiomes manipulated (endozoochory).
3. Collembola presence in permafrost will increase $CO_2$ production, through both collembola respiration and a
   stimulation of microbial activity ('priming effect').
4. The gradual introduction of distinct bacterial communities hypothesized above (2) will result in increased $CO_2$
   production, owing to the functional limitations of Yedoma permafrost microbial community in terms of $CO_2$
production.



## 2. Materials and Methods

### 2.1 Experimental design

#### 2.1.1 Soils

The Yedoma sediment used to assess the potential of the different microbial communities to alleviate functional limitations

originated from the Cold Region Research Engineering Laboratory (CRREL) Permafrost Tunnel (Fox, Alaska, USA). The sediment was sampled from the upper silt unit and is an upper Pleistocene silty deposit, previously described in details (Shur et al., 2004; Mackelprang et al., 2011, 2017; Monteux et al., 2020). This sediment was chosen due to its microbial functional limitations, allowing to discern impacts of introduced microorganisms on broad proxies such as $CO_2$ production (Monteux et al., 2020). Approximately 35 g (fresh weight) homogenized sediment was set in 200ml glass jars, sealed with parafilm to

allow for gas but not moisture or microorganism exchange, and pre-incubated at 10 °C for 11 days before inoculation.

To manipulate the collembola microbiome and make it more similar to that found in natural settings, we used a topsoil (0-15 cm depth) from a subarctic meadow (Kärkevagge, 30 km west of Abisko, northern Sweden, 68°24'23.8"N, 18°18'51.6"E) sampled in September 2019 and kept frozen until the cultivation of collembola.

#### 2.1.2 Collembola

A strain of the collembola species *Folsomia candida* was obtained from Vrije Universiteit Amsterdam and cultured 6 months prior to the onset of the experiment. *Folsomia candida* is a parthenogenetic ground-dwelling collembola, which has been routinely used as a model organism in soil ecology. Stock cultures were maintained on a gypsum and coal medium and fed baker's yeast, traces of mould were removed and fresh yeast and water were added once to twice a week, and fresh stock cultures were started monthly.

#### 2.1.3 Collembola inoculation treatments

Two months prior to the onset of the experiment, stock cultures were established on gypsum and coal medium supplemented with a 2-3 cm layer of topsoil, to obtain collembola which both skin and gut microbiome were colonized with topsoil microorganisms. In parallel, similar stock cultures using permafrost sediment were established as an additional control. These stock cultures with soil or sediment were supplemented with yeast and water alike the gypsum stock culture, to

maintain high adult population densities. One day before inoculating the incubation jars, soil suspensions were made from the topsoil and permafrost sediment (5 and 10 g, respectively, in 100 ml $ddH_2O$), shaken at 150 rpm for 1 hour and filtered (Ahlstrom-Munksjö grade 006, 1.5 µm pore size).

All incubation jars containing permafrost were randomly assigned a treatment on the day of inoculation among the

following:

- No collembola (hereafter, 'No-collembola control');



- Collembola from stock culture ('Collembola');

- Collembola sprayed with topsoil suspension ('Ectozoochory');

- Collembola sprayed with permafrost suspension ('Ectozoochory control');

- Collembola grown on topsoil stock culture ('Endozoochory');

- Collembola grown on permafrost stock culture ('Endozoochory control').

Adult collembola were isolated from the stock cultures using a hand-held vacuum cleaner mounted with a 10 ml pipet tip. The collembola were transferred into a black plastic tray, allowing to spread them and pick out the adults, excluding juveniles as much as possible. From this tray, a similar amount of collembola (30 – 80 individuals, in the range of values

used in the literature e.g. (Kaneda and Kaneko, 2008)) was sampled with the vacuum cleaner, then inoculated into each jar by pouring them into a plastic funnel, with the collembola provenance or manipulation depending on the treatment. Collembola for both endozoochory treatments were taken from their respective stock cultures, while those for the ectozoochory and stock collembola treatment were taken from the gypsum-coal-yeast stock culture. Prior to pouring them into the plastic funnel, collembola used in the ectozoochory treatment and its control were sprayed with the corresponding

soil suspensions. To limit cross-contamination, separate funnels, as well as separate 10 ml pipet tips on the vacuum cleaner were used for the different treatments. A picture of the inside of each jar was taken to count the exact number of collembola, then the jars were closed with rubber septa, flushed with moisturized $CO_2$-free air and incubated. To estimate collembola respiration per individual, collembola were also incubated in 8 jars under identical conditions, except that the soil was replaced with a few drops of autoclaved $ddH_2O$ to prevent dehydration (hereafter, no-soil calibration).

**2.1.4. Incubation**

We dark-incubated all flasks for 25 days under aerobic conditions at 10 °C. This incubation temperature is similar to summer active-layer temperatures in permafrost affected areas and within the thermal tolerance range of psychrophilic microorganisms (D'Amico et al., 2006). We used a short (25 days) incubation period to ensure a relatively stable collembola population level, by limiting uncertain numbers of newly-hatched collembola individuals, since the eggs of *Folsomia*

*candida* take 18-20 days to hatch at 16 °C (Marshall & Kevan, 1962) and presumably longer than 25 days at 10 °C.

**2.2. Measurements**

**2.2.1. CO₂ production**

Headspace air was sampled with a syringe to measure $CO_2$ concentrations (EGM-5 IRGA, PP Systems, Amesbury, Massachusetts, USA) at intervals ensuring $CO_2$ concentrations remained below 20,000 ppm to prevent a toxic $CO_2$ build-up

(i.e. after 3, 7, 14 and 25 days). After each measurement, the jars were flushed with 0.45 µm-filtered $CO_2$-free air moisturized by bubbling it through two 5l bottles of $ddH_2O$, for 3 min at 1 to 2 l min$^{-1}$, i.e. with at least 15 times the volume





of the jar. $CO_2$ concentrations were adjusted for changes in temperature and atmospheric pressure to calculate $CO_2$ production rates ($\tau$) as follows:

$$\tau_{(i)} = \frac{[CO_2]_i \times (P_i V / RT_i)}{(\Delta_t)_i}$$

where $(\Delta_t)_i$ is the time interval between measurement $_{(i)}$ and previous flushing, $P_i$ is atmospheric pressure at measurement time, V the headspace volume, R the ideal gas constant and $T_i$ the temperature. To calculate cumulative $CO_2$ production over the entire incubation, we summed up the quantity of $CO_2$ present in the headspace at each sampling, within each jar.

### 2.2.2. Bacterial community

Microcentrifuge tubes (1.5 ml) were filled with soil and snap-frozen in dry ice to analyse microbial communities from the
jars harvested after one day and at the end of the incubation. Initial soils were sampled likewise, after thawing and homogenizing but before pre-incubation for the permafrost soils and before preparing soil suspensions for active layer soils. The frozen tubes were kept at -20 °C for up to 4 months before freeze-drying, then homogenized by bead-beating (Precellys CK-68 15 ml tubes, $2 \times 30$ s at 4500 rpm). DNA was extracted from 183 to 285 mg of homogenized freeze-dried soil using DNEasy PowerSoil Pro Kit (Qiagen) according to the manufacturer's instructions, and DNA concentrations in the extracts
were measured on a Qubit 1.0 fluorometer.

The V4-V5 region of the 16S ribosomal RNA gene was targeted in PCR amplification using primers 515F (5' GTGYCAGCMGCCGCGGTAA 3') and 926R (5' CCGYCAATTYMTTTRAGTTT 3') with Illumina sequencing adapters, using 12.5 µl Phusion Taq Green PCR mastermix (Thermo Scientific), 0.25 µM of each primer, 2 µl of DNA extract diluted
to 5 ng µl$^{-1}$ and nuclease-free water to 25 µl reaction volume. PCR conditions were as follows: initial denaturation (98 °C, 3 min), 25 cycles of denaturation (98 °C, 15 s), annealing (50 °C, 30 s) and elongation (72 °C, 40 s), and a final elongation (72 °C, 10 min), after which PCR products were checked by electrophoresis on 1% agarose SB gel. 20 µl of PCR products were cleaned and their DNA concentrations normalized using a SequalPrep Normalization Plate kit (Invitrogen), according to manufacturer's instructions. Three DNA extraction blanks and two PCR blanks were included as negative controls, as
well as a mock community as positive control (ZymoBIOMICS, diluted to 5 and 0.5 ng µl$^{-1}$ in two replicates each).

A second PCR step was performed to add Nextera dual-indexing barcodes, using 30 µl reaction volume, 1 µM of each primer and 5 µl cleaned PCR product. PCR conditions were as follows: initial denaturation (98 °C, 3 min), 8 cycles of denaturation (98 °C, 30 s), annealing (55 °C, 30 s) and elongation (72 °C, 40 s), and a final elongation (72 °C, 10 min), after
which PCR products were checked by electrophoresis on 1% agarose SB gel. 25 µl of PCR products were cleaned and their DNA concentrations normalized using a SequalPrep Normalization Plate kit (Invitrogen). Serial elution across columns was used to increase concentration of the pooled products, i.e. using only $8 \times 20$ µl elution buffer instead of $96 \times 20$ µl. The eluted DNA was pooled, its concentration was measured on a Qubit fluorometer and the size distribution of the amplicons





was measured by automated electrophoresis (Agilent 2100 Bioanalyzer). The library was then sent for sequencing on an
Illumina MiSeq with V3 chemistry (2 × 300 bp, 15% PhiX spike-in) at the SNP&SEQ Technology Platform in Uppsala.
Demultiplexing was performed by the sequencing facility, and data deposited at ENA with accession number PRJEB51992.

### 2.3. Data analysis

### 2.3.1. Bioinformatics

All bioinformatics and statistics were performed in R v4.1.3 (R Core Team, 2019), unless specified otherwise. The whole
analysis pipeline is found at https://git.bolin.su.se/smonteux/Collembola_vector and the processed data and figure-generating
script at Zenodo (Monteux et al., 2022). In short, amplicon sequence variants (ASVs) were created with DADA2 1.18.0
(pseudo-pooling; Callahan et al., 2016) after removing primers and adapters with cutadapt (v3.10; Martin, 2011). Taxonomy
was assigned to ASVs with the RDP naïve Bayesian classifier (v1.8; Wang et al., 2007), and ASVs resolved to the genus
rank were further assigned a species rank by the exact string-matching algorithm implemented in DADA2 (*assignSpecies*),
using SILVA v138.1 reference data (Quast et al., 2013). Putative contaminant ASVs were manually selected from those
identified *in silico* using the *decontam* algorithm (Davis et al., 2018) with combined prevalence- and frequency-based
methods using the default threshold of 0.1, and separate prevalence- and frequency-based methods with a threshold of 0.05.
Eight contaminant ASVs amounting up to 0.024% of the total reads were removed. Appropriateness of the bioinformatics
analysis parameters was judged by visually assessing the composition of mock communities (microbial community DNA
standard, ZymoBIOMICS) at the genus level, leading to ASVs amounting up to less than 10 reads and/or present in fewer
than 3 samples being removed from the dataset. Sequencing depth did not associate with experimental treatments (ANOVA
$F_{5,30}$=0.701, $P$=0.627), therefore read numbers were converted to proportional abundances within samples to normalize
sample sizes.

### 2.3.2. Diversity analyses

The effect of the different treatments on bacterial communities was visualized using principal coordinates analysis (PCoA)
and tested with permutational multivariate analyses of variance (PerMANOVA, *adonis* function in vegan package) after
verifying homoscedasticity (*betadisper*). Two distance matrixes were computed for that purpose, using weighted- and non-
weighted UniFrac distances (Lozupone et al., 2011), to distinguish between compositional effects accounting for bacterial
relative abundances and for only presence-absence, respectively. Pairwise contrasts were subsequently computed using the
wrapper provided in the pairwiseAdonis R package (Arbizu, 2021).
In addition, the same effect was tested with the more robust and sensitive *manyGLM* approach (Wang et al., 2012), which
avoids certain pitfalls of distance-based methods (Warton et al., 2012). ManyGLM were fitted with negative binomial
distribution, after visually checking that assumptions were met, on the non-normalized ASV count data using the default
PIT-trap resampling with 1999 bootstrap permutations, and a likelihood-ratio testing method. An analysis of deviance was



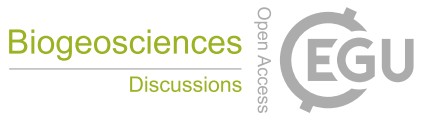

carried out (*anova.manyglm*), subsequently using the provided wrapper for pairwise comparisons with free step-down *P* value adjustment.

### 2.3.3. CO$_2$ production analysis

A repeated-measures ANOVA was carried out, using Greenhouse-Geisser epsilon correction to degrees of freedom to account for the violation of the sphericity assumption, to assess the interactive effect of the collembola manipulation
treatments over time. Since this interactive effect was not statistically significant ($P$=0.56, Appendix A1), we removed the time dimension and used cumulative CO$_2$ production at the end of the incubation period below.

To assess our hypothesis that collembola presence would increase CO$_2$ production we used two-sample t-tests with unequal variances to compare the cumulative CO$_2$ production at the end of the incubation between the no-collembola control and all other jars.

To explore differences between the collembola microbiome manipulations, we used a one-way ANOVA followed by treatment contrasts to assess the difference from the control (i.e. all treatment and controls compared to the no-collembola control) and selected orthogonal contrasts to assess the effects of each treatment (i.e. each treatment compared to its own control).

### 2.3.4. Estimated collembola respiration

Based on photographs taken upon inoculating the incubation jars with collembola and three days after inoculating, we counted the exact number of live collembola (i.e. ignoring collembola which had apparently not moved between the two pictures were taken). Only a few animals did not survive the transfer, and no dead animals were observed at the end of the incubation. The amount of collembola per jar varied across treatments ($F_{4,25} = 3.19$, $P = 0.030$), although no pairs of treated jars significantly differed from each other (when including the no-soil calibration jars, $F_{5,30} = 4.048$, $P = 0.006$; Appendix B),
we therefore needed to account for differing collembola numbers. We averaged the respiration per individual in the no-soil calibration set for each of the CO$_2$ concentration measurement times, and multiplied this amount by the number of collembola present in each jar to estimate the amount of CO$_2$ produced by collembola basal respiration in treatment jars.

### 2.3.5. Response-ratio calculations

A response-ratio of CO$_2$ production in collembola treatments was calculated by dividing cumulative CO$_2$ production by the
average of the no-collembola control. This was done both for the net CO$_2$ production after subtracting the estimated collembola basal respiration (RR$_{soil}$), and for the gross CO$_2$ production (RR$_{gross}$). This allows to partition the difference in altered CO$_2$ production between what was respired by the collembola and a putative priming effect on soil CO$_2$ production.





# 3. Results

## 3.1. Bacterial communities

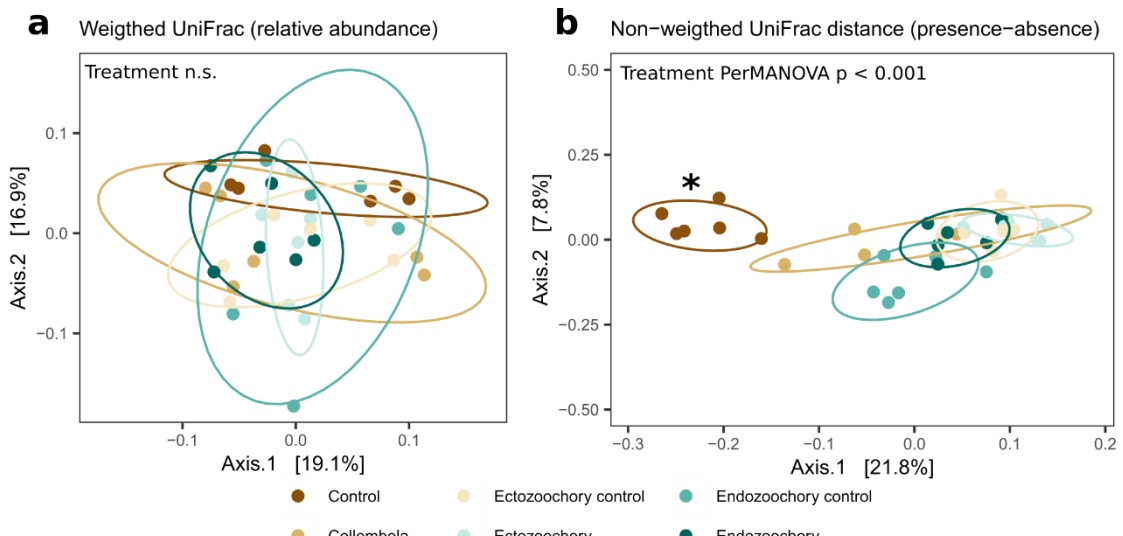

**Figure 1: Response of permafrost bacterial communities to the addition of collembola undergoing different microbiome manipulation**. **a:** Principal coordinates analysis (PCoA) of normalized abundance of bacterial ASVs based on weighted UniFrac phylogenetic distance (i.e. accounting for relative abundances); **b:** PCoA of normalized abundance of bacterial ASVs based on non-weighted UniFrac phylogenetic distance (i.e. presence-absence only). Ellipses indicate 80% confidence interval around treatment centroids, and axes are scaled to the percentage of explained variance. The asterisk indicates that the No-collembola control (dark brown) in **b** significantly differs from all other treatments (pairwise PerMANOVA *p<0.05*) and collembola treatments did not differ from the stock-collembola treatment. Pairwise comparisons are summarized in Appendix C1.


Bacterial community composition was overall largely unaffected by the presence of collembola. Taking relative abundances into account, no effect of the different treatments was identified with PERMANOVA ($P > 0.05$, Figure 1a), but a significant treatment effect was observed using the presence-absence distance metrics (PERMANOVA $F_{5,30}$=2.91, $P = 0.001$, Figure 1b). More specifically, treatments containing collembola differed from the no-collembola control (Figure 1b) and to some

extent from each other (Appendix C1). Using manyGLM, a significant treatment effect was observed (manyGLM analysis of deviance $P < 0.001$; Table 1). The ectozoochory treatment and its control (collembola sprayed with topsoil or permafrost soil suspension, respectively) differed significantly from the no-collembola control in the post-hoc test ($0.01 < P < 0.05$), while the other treatments including collembola tended to differ from the no-collembola control ($0.05 < P < 0.1$, Table 1). Similarly to the presence-absence analysis, treatments that included collembola did not significantly differ from each other in

the manyGLM analysis. Overall, soil with collembola exhibited higher alpha-diversity metrics (richness estimators Chao1, ACE, number of observed ASVs, as well as Shannon and Fisher diversity indexes, Figure 2, Appendix C2) than the no-collembola control but those did not differ among the collembola treatments.



**Table 1: Effect of addition of collembola on Yedoma sediment bacterial communities.**
Analysis of deviance test on a manyGLM model fitted on 1206 ASVs assuming negative-binomial distribution, using 1999 iterations of PIT-trap resampling.
$P$ values below 0.05 are denoted in bold; ***: $P < 0.001$; *: $0.01 < P < 0.05$; .: $0.05 < P < 0.1$.

| Multivariate test | | | Res.Df | Df.diff | Dev | P | |
|---|---|---|---|---|---|---|---|
| | Treatment | | 30 | 5 | 11742 | **<0.001** | *** |

| Multivariate test contrasts | | | Observed statistic | Adjusted P | |
|---|---|---|---|---|---|
| Control | vs | Collembola | 3124 | 0.082 | . |
| Control | vs | Ectozoochory control | 4085 | **0.013** | * |
| Control | vs | Ectozoochory | 4274 | **0.011** | * |
| Control | vs | Endozoochory control | 3276 | 0.064 | . |
| Control | vs | Endozoochory | 3117 | 0.082 | . |
| Collembola | vs | Ectozoochory control | 1533 | 0.669 | |
| Collembola | vs | Ectozoochory | 1580 | 0.669 | |
| Collembola | vs | Endozoochory control | 1842 | 0.658 | |
| Collembola | vs | Endozoochory | 1720 | 0.669 | |
| Ectozoochory control | vs | Ectozoochory | 1755 | 0.669 | |
| Ectozoochory control | vs | Endozoochory control | 2335 | 0.378 | |
| Ectozoochory control | vs | Endozoochory | 1752 | 0.669 | |
| Ectozoochory | vs | Endozoochory control | 2375 | 0.378 | |
| Ectozoochory | vs | Endozoochory | 2031 | 0.577 | |
| Endozoochory control | vs | Endozoochory | 1976 | 0.586 | |


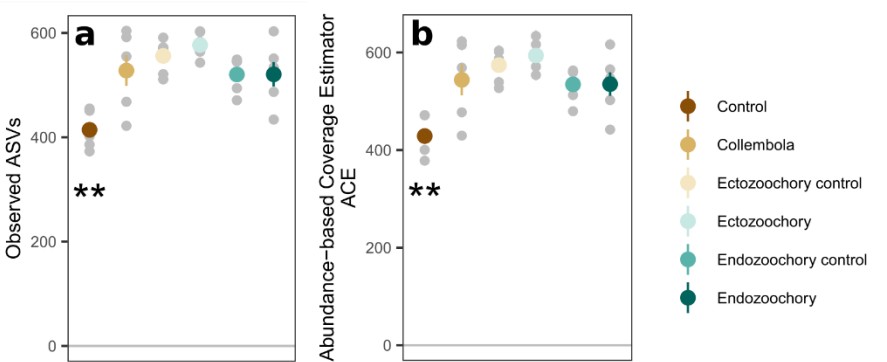

**Figure 2: Response of permafrost bacterial alpha diversity to the addition of collembola undergoing different microbiome manipulation**. **a:** Number of observed ASVs per sample; **b:** Abundance-based Coverage Estimator ACE. Coloured symbols and error-bars are means and standard-errors within a treatment (n=6), small grey symbols are individual samples. Error-bars are only shown when exceeding the symbols size. Asterisks denote that the no-collembola control differs from all other treatments (estimated marginal means pairwise comparison, 0.001 < Holm-adjusted $P$ < 0.01).




## 3.2. CO₂ production

Overall, the collembola addition resulted in higher $CO_2$ production that the no-collembola control, but the ectozoochory and
endozoochory treatments did not result in higher $CO_2$ production than their respective controls (using gross $CO_2$ production, as in $RR_{gross}$). When accounting for collembola basal respiration (as in $RR_{soil}$), only the ectozoochory treatment (p=0.005) and to a lesser extent its control (p=0.069) differed from the no-collembola control (Appendix A2).

We hypothesized an increased $CO_2$ production in the zoochory treatments through an effect on bacterial communities, however that effect was absent, we therefore tested for the overall effect of collembola presence across all treatments compared to the no-collembola control (Figure 3). Using a t-test with unequal variances, we observed a 25.85% increase in $CO_2$ production in presence of collembola ($RR_{gross}$; Figure 3a). When subtracting estimated collembola basal respiration, this increase was on average 13.22%, ($RR_{soil}$; Figure 3b), thus roughly half of the observed increase in $CO_2$ production could be attributed to collembola basal respiration.

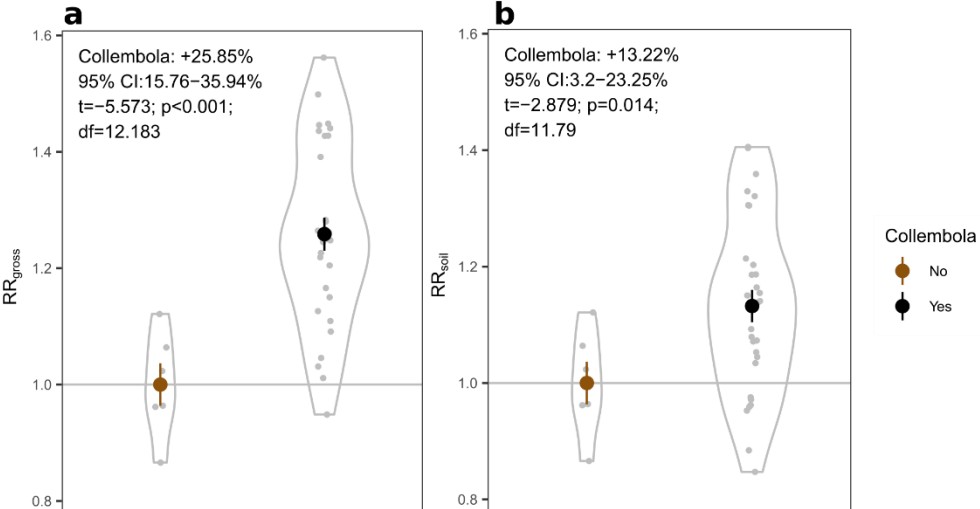

**Figure 3: Response-ratios of permafrost cumulative $CO_2$ production rates to addition of collembola**. **a:** $RR_{gross}$: measured $CO_2$ production rates; **b:** $RR_{soil}$: measured $CO_2$ production rates with estimated collembola basal respiration subtracted. All values are divided by the mean of the no-collembola control, small grey circles are individual values, large symbols are means, error-bars are standard errors of the mean (n=6 and 30 for brown and black symbols, respectively), and violin plots denote the shape of the distributions with maximum width proportional to the number of observations. Statistics are derived from two-sample t-tests with unequal variances.

## 4. Discussion

We showed that while collembola seem to stimulate $CO_2$ production from permafrost sediment through 'priming', their effect as vectors of dispersal for bacteria seems minimal compared to the effects of their simple presence on bacterial communities. Such effects may be more visible on longer timescales or on other components of the soil food web.



### 4.1. Introduction of bacteria by collembola

Our hypothesis that collembola presence would modify permafrost bacterial community composition, was not clearly supported or rejected. The presence of collembola did not induce significant changes in bacterial community composition as a whole (i.e. when considering relative abundances, Figure 1a, Table 1). However, collembola did introduce new bacteria, resulting in an increase in alpha-diversity and significant differences in bacterial community composition on a presence-absence basis, but these introduced microbes did not depend on the provenance/treatment assigned to the collembola (Figure
1b, Figure 2).

### 4.2. Differences between ecto- and endo-zoochory treatments

Contrary to our expectations, our hypothesis stating that collembola manipulated to permit both ecto- and endo-zoochory would result in a different bacterial community composition in Endozoochory samples than in those allowing for Ectozoochory alone was rejected. It may be that our method does not alter the cuticle microbiome: collembola cuticle is
particularly hydrophobic (Hensel et al., 2013a, b; Nickerl et al., 2013) and that might have impeded adherence of bacteria to the cuticle upon spraying. This should however not be such an issue for the Endozoochory treatment, where collembola were in contact with the topsoil for a prolonged (>1 month) period of time. Despite this prolonged exposure, no additional effect of Endozoochory on bacterial community composition was observed, compared to Ectozoochory or Collembola samples. The presence of bacteria from the topsoil in the collembola gut was very likely, considering the piles of soil-coloured feces
in the stock cultures, as collembola typically harbor a gut microbiome reflecting their diet and direct environment (Xiang et al., 2019; Leo et al., 2021). In contrast with the permafrost stock cultures, where the yeast was necessary as an additional food source to maintain stable populations, collembola populations remained stable in topsoil stock cultures similar to those used in the Endozoochory treatment even in absence of yeast, further indicating their consumption of soil particles and associated microorganisms. Therefore, the absence of effect of both Ecto- and Endo-zoochory treatments on permafrost
bacterial communities may indicate that collembola cuticle does not serve as a vector for bacterial dispersal, possibly due to its omniphobic structure. Further, the lack of difference between the effects of Collembola and Endozoochory treatments suggests that few to none of the bacteria present in this topsoil are able to establish in- or survive transit through collembolan gut. This suggests against generalist 'hitch-hiking' bacteria using microfauna guts as a means of dispersal, emphasizing instead that such hitch-hiking bacteria are similar regardless of the feeding context of the collembola.


We cannot rule out that the observed limited response of bacterial community to zoochory treatments was transient, and that the collembola presence would have eventually affected community composition also in terms of relative abundances. A longer incubation period may thus have resulted in stronger effects on bacterial community composition, as observed by Coulibaly et al., (2019), but this would have been at the expense of controlling the number of collembola in the jars, and thus
of being able to account for their basal respiration. Using a RNA-based approach to target the 'active' bacterial community

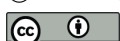



may be another way to detect such putative transient effects. It should also be noted that our semi-quantitative approach to bacterial community composition did not allow us to assess putative changes in absolute abundances, although such changes are not necessarily observed in similar studies (Kaneda and Kaneko, 2008).

Fungal communities may have been more responsive to the presence of mostly fungivorous *Folsomia candida*, however fungi are often scarce in newly-thawed permafrost. Our earlier work on this sediment suggested its fungal community was dominated by two fungal strains, and the use of *Candida albicans* as food source for the collembola stock cultures may have strongly distorted fungal community composition, thus requiring questionable bioinformatic workarounds. Further, assessing the effects on fungal communities may not allow to distinguish between the consequences of collembola as a vector for

microbial migration and as a grazing consumer. Taken together with the preponderant role of bacterial groups on realizing soil functions in this system (Monteux et al., 2020), this explains our choice of bacterial community as a response variable.

### 4.3. Collembola effects on $CO_2$ production

Our hypothesis stating that soil $CO_2$ production would be increased in presence of collembola was supported by our results. However, since no differences in bacterial community composition was detected between the different collembola

treatments, the rationale for our follow-up hypothesis (attributing an increased influence on $CO_2$ production to the treatments exhibiting stronger changes in bacterial community) was not met. We therefore did not deem relevant to formally evaluate this hypothesis. Our results suggest that it would be rejected (Appendix A3), but we could not say whether that is because the rationale is wrong or because the manipulation of bacterial communities did not yield the expected effect.

Overall, collembola presence increased respiration by 25%, half of which was attributed to respiration of collembola themselves. We interpret the remaining 13% as a "priming effect", i.e. a stimulation of SOM-derived $CO_2$ production induced by the presence of collembola. The mechanism behind this priming effect remain unclear, and could for instance relate to stimulation of microbial activity by the input of more labile substrates by the collembola (digestion by-products). Overall, our results add to a list of contrasting findings on the direction and magnitude of priming of SOM decomposition by

collembola presence in soils in general. Earlier studies that compared $CO_2$ production in presence or absence of collembola or other faunal groups have mostly focused on litter, e.g. Hanlon and Anderson, (1979). The scarce studies on soil $CO_2$ production including collembola reported contrasted findings, with some studies observing increased soil $CO_2$ emissions in presence of collembola (+0% to +400% in Addison and Parkinson (1978), +16% in Bakonyi (1989), up to +27% in Kaneda and Kaneko (2008), +25% in Wang et al. (2017)), while others found no significant changes (Theenhaus et al., 1999;

Lubbers et al., 2020; Lucas et al., 2020). Collembola species identity has been found to significantly affect their effect on $CO_2$ production in litter (Hanlon and Anderson, 1979), and our results could support this as we found a similar increase as Bakonyi (1989), Kaneda and Kaneko (2008), and Wang et al. (2017), three studies where *Folsomia candida* was used as well. Kaneda and Kaneko (2008) reported a density-dependent increase in $CO_2$ production with the addition of *F. candida*,

which showed no effect at the density used in our study but an increase with 5-10 times higher density (i.e. 400 individuals
in 30g soil compared to our 30-80). Accordingly, we did not find any trend supporting a density-dependent response in our
data. Nevertheless, we suggest that permafrost soils colonized by collembola in particular, and soil fauna in general, will
likely see an increase in their rates of decomposition, and $CO_2$ production in particular, due to both priming and the basal
respiration of the newly-established food web.

Our experimental setup did not allow us to distinguish further between soil and collembola respiration. We estimated basal
respiration, though this may not perfectly reflect reality, considering that collembola in jars without soil may not have
behaved similarly to those in jars with soil. Faced with a scarcity of food sources, they may have slowed their metabolism to
endure that stress, which could result in decreased basal respiration, thus leading us to overestimate the magnitude of
"priming". We could not find reference values in the literature for *Folsomia candida* respiration rates, but Addison &
Parkinson (1978) estimated the $CO_2$ production of two high-Arctic collembolan species to 182.6 and 250.6 µl $CO_2$ per g live
weight per hour at 10 °C, for *Hypogastrura tullbergi* and *Folsomia regularis*, respectively. Assuming a $CO_2$ density of 1.977
kg / m3 and 170 µg per adult individual of *Folsomia candida*, our values range between 47 and 220 µl $CO_2$ per live weight
per hour. Addison and Parkinson (1978) refer to Mitchell (1973) for a description of the method used for estimating
respiration rates, where the measurements appear to be carried over a period of 20 hours. When considering only the first
measurement date in our data (3 days after inoculation), the rates are closer to the values reported by Addison and Parkinson
(1978) with 157.5 ± 18.5 µl $CO_2$ per live weight per hour (mean ± SE, n=6). In future studies of this mechanism, it would be
interesting to use stable isotope methods to partition the collembola- vs soil organic matter-derived $CO_2$ production more
accurately.

Beyond that due to soil fauna, our collembola addition treatments may have induced a "microbial community priming", in
the form of the release of microbial functional limitations. The Yedoma sediment used in this study indeed lacks certain
microbial taxa and functions, and the onset of an exotic microbial community can result in large increases in $CO_2$ production
(Monteux et al., 2020). Considering the limited to non-significant (depending on the distance metrics) effect of collembola
addition on bacterial communities, we do not think this mechanism explains our observed results, although we do not rule
out this explanation.

## 5. Conclusions

Collembola presence modified bacterial communities in newly-thawed permafrost and primed its $CO_2$ production. An
emerging theme in permafrost research is the missing functions in permafrost microbial communities due to a lack of certain
microbial groups, hampering the production of $CO_2$, $CH_4$ and $N_2O$ (Knoblauch et al., 2018; Monteux et al., 2020;
Marushchak et al., 2021). It is also becoming clearer that the functionality of permafrost microbes can vary across space





(Barbato et al., 2022), therefore the modalities of microbial dispersal into newly-thawed permafrost likely affect the fate of its organic matter and the rate of release of greenhouse gases. Our findings suggest that collembola may accelerate the release of greenhouse gases, possibly in part through the introduction of microorganisms, although those did not seem to depend strongly on the preceding collembola environment. In nature, thawing permafrost mostly occurs at the bottom of the

seasonally-thawing active layer, often below the water-table and at depths far from the topsoil where most collembola reside. Different soil fauna, such as earthworms, may access such newly-thawed permafrost but their current geographical range does not overlap much with the permafrost region. However, rapid thaw events such as hillslope thermokarst, active layer detachments, or retrogressive thaw slumps expose newly-thawed permafrost to surface conditions, thus making it susceptible to colonization by collembola. Fauna-induced dispersal of microorganisms is most likely to take place in these environments,

and should be further investigated in realistic field conditions.

It remains unclear whether the increase in $CO_2$ production we attribute to priming stems from increased functionality of the microbial community due to bridging of functional limitations, or from other mechanisms. Mechanical breakdown of organic material and digestion processes might provide microorganisms with more easily available substrates, while disruption of the

soil pore structure may also result in increased microbial decomposition activity. Several studies indicate increases in $CO_2$ production with the presence of soil fauna, however to our best knowledge no studies have specifically partitioned to which extent such increases were due to faunal respiration or to priming effects. We argue that elucidating this question would be an important next step towards opening the 'black box' that soil systems still often represent, thus helping to mechanistically address the effects of global changes.





**6. Appendices**

**Appendix A1: Effect of collembola addition on daily $CO_2$ production over time, repeated-measure ANOVA.**

*P* [GG] indicates RM-ANOVA *P* using Greenhouse-Geisser correction on degrees of freedom to account for violation of the assumption of sphericity (Mauchly's test).

P values below 0.05 are denoted in bold; ***: P < 0.001; *: 0.01 < P < 0.05.

| RM-ANOVA | DFnum | DFden | F | P | ges | |
|---|---|---|---|---|---|---|
| Treatment | 5 | 30 | 3.499 | **0.013** | 0.259 | * |
| Date | 3 | 90 | 65.263 | **<0.001** | 0.465 | *** |
| Treatment:Date | 15 | 90 | 0.854 | 0.616 | 0.054 | |

| Mauchly's sphericity test | W | P | |
|---|---|---|---|
| Date | 0.165 | **<0.001** | *** |
| Treatment:Date | 0.165 | **<0.001** | *** |

| Sphericity corrections | GGe | p[GG] | |
|---|---|---|---|
| Date | 0.544 | **<0.001** | *** |
| Treatment:Date | 0.544 | 0.562 | |





**Appendix A2: Effects of collembola additions on response ratios of cumulative CO$_2$ production at the end of the incubation**, excluding (RRsoil) or including (RRgross) estimated collembola respiration. Holm adjustment for $P$ values of non-orthogonal contrasts.

P values below 0.05 are denoted in bold; ***: $P < 0.001$; **: $0.001 < P < 0.01$; *: $0.01 < P < 0.05$; .: $0.05 < P < 0.1$.

**RR$_{soil}$**

| ANOVA | Df | Df.Res | Sum Sq | Mean Sq | F value | P | |
|---|---|---|---|---|---|---|---|
| Treatment | 5 | 30 | 0.003 | 0.001 | 3.728 | **0.010** | * |

| Treatment contrast (vs Control) | Estimate | SE | Df | t.ratio | Adjusted P | |
|---|---|---|---|---|---|---|
| Collembola | 0.012 | 0.008 | 30 | 1.573 | 0.379 | |
| Ectozoochory control | 0.020 | 0.008 | 30 | 2.525 | 0.068 | . |
| Ectozoochory | 0.028 | 0.008 | 30 | 3.626 | **0.005** | ** |
| Endozoochory control | 0.003 | 0.008 | 30 | 0.443 | 0.764 | |
| Endozoochory | 0.007 | 0.008 | 30 | 0.887 | 0.764 | |

| Orthogonal contrasts | Estimate | Std. Error | t value | P |
|---|---|---|---|---|
| Collembola vs Control | 0.006 | 0.004 | 1.573 | 0.126 |
| Ectozoochory vs Ectozoochory control | 0.002 | 0.004 | 0.444 | 0.660 |
| Endozoochory vs Endozoochory Control | 0.004 | 0.004 | 1.101 | 0.280 |

**RR$_{gross}$**

| ANOVA | Df | Df.Res | Sum Sq | Mean Sq | F value | P | |
|---|---|---|---|---|---|---|---|
| Treatment | 5 | 30 | 0.005 | 0.001 | 4.184 | **0.005** | ** |

| Treatment contrast (vs Control) | Estimate | SE | Df | t.ratio | Adjusted P | |
|---|---|---|---|---|---|---|
| Collembola | 0.024 | 0.009 | 30 | 2.754 | **0.030** | * |
| Ectozoochory control | 0.030 | 0.009 | 30 | 3.362 | **0.009** | ** |
| Ectozoochory | 0.038 | 0.009 | 30 | 4.331 | **0.001** | *** |
| Endozoochory control | 0.020 | 0.009 | 30 | 2.270 | **0.031** | * |
| Endozoochory | 0.023 | 0.009 | 30 | 2.629 | **0.030** | * |

| Orthogonal contrasts | Estimate | Std. Error | t value | P | |
|---|---|---|---|---|---|
| Collembola vs Control | 0.012 | 0.004 | 2.754 | **0.010** | ** |
| Ectozoochory vs Ectozoochory control | 0.002 | 0.004 | 0.359 | 0.722 | |
| Endozoochory vs Endozoochory Control | 0.004 | 0.004 | 0.969 | 0.340 | |

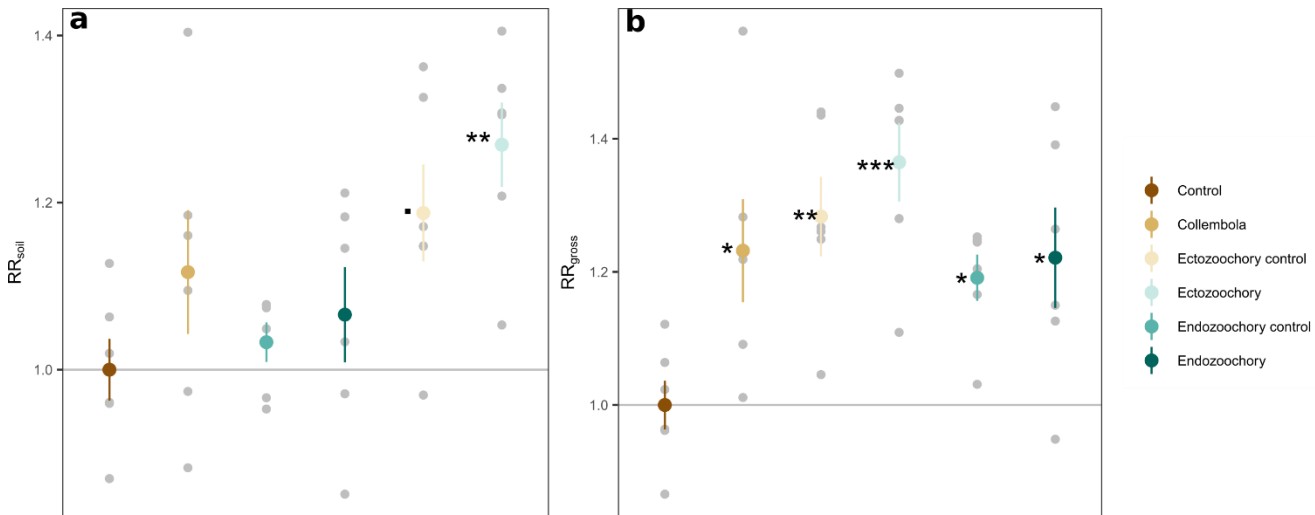

**Appendix A3: Response-ratios of permafrost cumulative CO$_2$ production rates to addition of collembola**. **a:** RR$_{gross}$: measured CO$_2$ production rates; **b:** RR$_{soil}$: measured CO$_2$ production rates with estimated collembola basal respiration subtracted. All values are divided by the mean of the no-collembola control, small grey circles are individual jar values, large symbols are means, error-bars are standard errors of the mean (n=6). Black symbols denote significant difference from the control, different symbols denote different statistical significance (EMmeans treatment contrasts with Holm adjustment for multiple comparisons, Supplementary Table 2, **.**: $0.05 < P < 0.1$; **\***: $0.01 < P < 0.05$; **\*\***: $0.001 < P < 0.01$; **\*\*\***: $P < 0.001$).

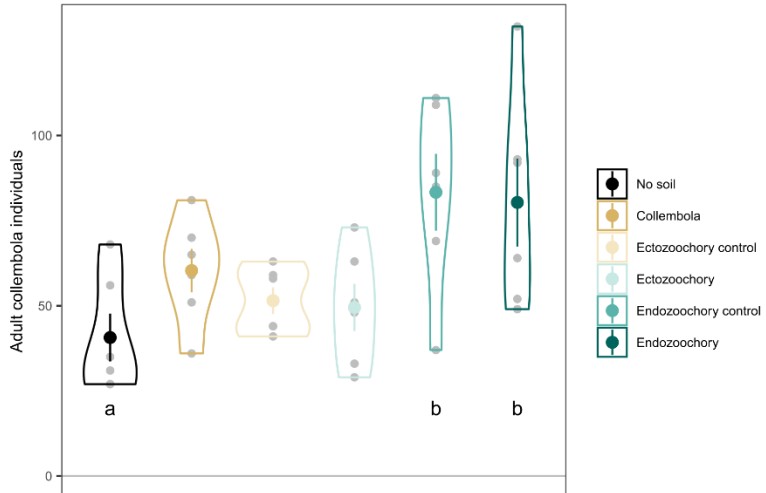

**Appendix B: Number of adult collembola individuals per incubation jar**. Small grey circles are individual jar values, large symbols are means, error-bars are standard errors of the mean (n=6), and violin plots denote the shape of the distributions with maximum width proportional to the number of observations. Different black letters denote statistically significant pairwise differences (EMmeans pairwise contrasts with Holm adjustment for multiple comparisons, $0.05 < P < 0.1$).





**Appendix C1: Effects of collembola additions on bacterial community composition (permutational multivariate ANOVAs and pairwise contrasts)**, using distance matrixes based on relative abundances (weigthed UniFrac) or presence-absence (non-weighted UniFrac). Holm adjustment for $P$ values of pairwise contrasts.

P values below 0.05 are denoted in bold; ***: $P < 0.001$; **: $0.001 < P < 0.01$; *: $0.01 < P < 0.05$; .: $0.05 < P < 0.1$.

| PerMANOVA | Df | Df.Res | F.Model | R2 | P |
|---|---|---|---|---|---|
| Weighted UniFrac (relative abundance) | 5 | 30 | 1.261 | 0.174 | 0.101 |
| Non-weighted UniFrac (presence-absence) | 5 | 30 | 3.018 | 0.335 | **< 0.001** |

| Pairwise contrast | Df | F.Model | R2 | Adjusted P | |
|---|---|---|---|---|---|
| Control vs Collembola | 1 | 3.236 | 0.244 | **0.025** | * |
| Control vs Endozoochory control | 1 | 4.356 | 0.303 | **0.025** | * |
| Control vs Ectozoochory | 1 | 6.777 | 0.404 | **0.022** | * |
| Control vs Ectozoochory control | 1 | 5.658 | 0.361 | **0.025** | * |
| Control vs Endozoochory | 1 | 4.777 | 0.323 | **0.012** | * |
| Collembola vs Endozoochory control | 1 | 1.767 | 0.150 | 0.102 | |
| Collembola vs Ectozoochory | 1 | 1.751 | 0.149 | 0.115 | |
| Collembola vs Ectozoochory control | 1 | 1.348 | 0.119 | 0.148 | |
| Collembola vs Endozoochory | 1 | 1.434 | 0.125 | 0.148 | |
| Endozoochory control vs Ectozoochory | 1 | 3.068 | 0.235 | **0.025** | * |
| Endozoochory control vs Ectozoochory control | 1 | 2.711 | 0.213 | **0.025** | * |
| Endozoochory control vs Endozoochory | 1 | 2.156 | 0.177 | **0.025** | * |
| Ectozoochory vs Ectozoochory control | 1 | 1.497 | 0.130 | 0.113 | |
| Ectozoochory vs Endozoochory | 1 | 1.958 | 0.164 | **0.025** | * |
| Ectozoochory control vs Endozoochory | 1 | 1.820 | 0.154 | **0.025** | * |






**Appendix C2: Effects of collembola additions on bacterial alpha diversity (ANOVAs and pairwise contrasts)**, with total number of observed ASVs, Abundance-based Coverage Estimator (ACE), Shannon's and Fisher's diversity indexes. Holm adjustment for $P$ values of pairwise contrasts.

P values below 0.05 are denoted in bold; ***: $P < 0.001$; **: $0.001 < P < 0.01$; *: $0.01 < P < 0.05$; .: $0.05 < P < 0.1$.

| ANOVA | Df | Df.Res | Sum Sq | Mean Sq | F value | P | |
|---|---|---|---|---|---|---|---|
| Observed ASVs | 5 | 30 | 94389 | 18878 | 9.313 | **<0.001** | *** |
| ACE | 5 | 30 | 98265 | 19653 | 8.517 | **<0.001** | *** |
| Shannon | 5 | 30 | 0.582 | 0.116 | 4.667 | **0.003** | ** |
| Fisher | 5 | 30 | 2141 | 428 | 12.644 | **<0.001** | *** |

| Treatment contrast (vs Control) | | Estimate | SE | df | t.ratio | P | |
|---|---|---|---|---|---|---|---|
| | Collembola | 113.500 | 25.994 | 30 | 4.366 | **0.001** | ** |
| | Ectozoochory control | 141.667 | 25.994 | 30 | 5.450 | **<0.001** | *** |
| Observed ASVs | Ectozoochory | 162.333 | 25.994 | 30 | 6.245 | **<0.001** | *** |
| | Endozoochory control | 106.000 | 25.994 | 30 | 4.078 | **0.001** | ** |
| | Endozoochory | 106.333 | 25.994 | 30 | 4.091 | **0.001** | ** |
| | Collembola | 115.2 | 27.7 | 30 | 4.155 | **0.001** | ** |
| | Ectozoochory control | 145.4 | 27.7 | 30 | 5.245 | **<0.001** | *** |
| ACE | Ectozoochory | 165.2 | 27.7 | 30 | 5.956 | **<0.001** | *** |
| | Endozoochory control | 106.0 | 27.7 | 30 | 3.821 | **0.003** | ** |
| | Endozoochory | 106.5 | 27.7 | 30 | 3.840 | **0.003** | ** |
| | Collembola | 0.230 | 0.091 | 30 | 2.521 | 0.070 | . |
| | Ectozoochory control | 0.326 | 0.091 | 30 | 3.571 | **0.006** | ** |
| Shannon | Ectozoochory | 0.378 | 0.091 | 30 | 4.148 | **0.001** | ** |
| | Endozoochory control | 0.198 | 0.091 | 30 | 2.171 | 0.144 | |
| | Endozoochory | 0.347 | 0.091 | 30 | 3.800 | **0.003** | ** |
| | Collembola | 16.806 | 3.360 | 30 | 5.002 | **<0.001** | *** |
| | Ectozoochory control | 21.731 | 3.360 | 30 | 6.468 | **<0.001** | *** |
| Fisher | Ectozoochory | 24.006 | 3.360 | 30 | 7.145 | **<0.001** | *** |
| | Endozoochory control | 14.598 | 3.360 | 30 | 4.345 | **0.001** | ** |
| | Endozoochory | 17.301 | 3.360 | 30 | 5.149 | **<0.001** | *** |
## 7. Code availability

All code used to process the raw DNA data and produce the figures and tables presented in text is found at
https://git.bolin.su.se/smonteux/Collembola_vector

The code used to generate figures from the processed DNA data is also found at https://dx.doi.org/10.5281/zenodo.6461323

## 8. Data availability

All 16S sequencing data is found at ENA under project accession number PRJEB51992.

All processed data used to generate the findings presented in text are found at https://dx.doi.org/10.5281/zenodo.6461323
and https://git.bolin.su.se/smonteux/Collembola_vector

## 9. Sample availability

Due to practical constraints, the exact permafrost material and collembola cultures used throughout the manuscript are no longer available. Permafrost from the same location may be obtained through contacting the USA Army CRREL, while *Folsomia candida* strains may be obtained by contacting JM. Frozen permafrost samples from the end of the incubation
period, as well as aliquots of DNA extracts used for this study may be obtained from the corresponding author upon reasonable request.

## 10. Author contributions

SM, JM and EK designed the experiment. JM provided *Folsomia candida* strains and guidance for culturing. EK provided topsoil from Kärkevagge. SM performed the experiment, data analysis, and wrote the manuscript with input from all co-
authors.

## 11. Competing interests

The authors declare that they have no conflict of interest.

## 12. Acknowledgements

This study was funded by a grant from Formas (Dnr 2017-01182) awarded to E.J.K. and Kempestiftelserna awarded to S.M.
We thank the Department of Forest Mycology and Plant Pathology, SLU, for hosting the molecular work, T. H. Douglas from the US Army Cold Regions Research and Engineering Laboratory's Permafrost Tunnel (Alaska) for assistance and



permission to sample. We also thank F. Keuper from the French National Research Institute for Agriculture, Food and Environment and E. Dorrepaal from Umeå University for providing the permafrost sample.

Sequencing was performed by the SNP&SEQ Technology Platform in Uppsala. The facility is part of the National Genomics
Infrastructure (NGI) Sweden and Science for Life Laboratory. The SNP&SEQ Platform is also supported by the Swedish Research Council and the Knut and Alice Wallenberg Foundation.

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
