# Peer review of "Dispersal of bacteria and stimulation of permafrost decomposition by Collembola"

_Biogeosciences, 2022_

## Author Comment (AC3)

[Figure]

**Figure 1: Summary of the experimental design.** Folsomia candida collembola were grown in a stock culture on gypsum and fed with yeast, then subjected to different treatments before being inoculated into jars filled with Yedoma permafrost sediment (30-80 individuals per jar). Ectozoochory was assessed by spraying the collembola with soil suspensions, while endozoochory was assessed by growing collembola in stock cultures supplemented with soil. The jars were incubated in the dark for 25 days at 10 °C, $CO_2$ production was measured throughout the incubation period and bacterial community composition was determined at the end of the incubation.

[Figure]

**AC Figure 1: Yedoma permafrost bacterial community composition with and without bacterial community manipulation** by inoculating with a soil suspension, after 389 days of dark aerobic incubation at 10 °C. Principal Coordinates Analysis of weighted UniFrac distances derived from ASVs obtained from amplicons of the 16S V4V5 regions, following laboratory and bioinformatic analyses similar to those described in the main text.

---

## Author Response (AR1)

**Author's response letter to review of bg-2022-98**

Dear Editor,

We have now addressed the comments raised during the interactive discussion phase and editors' suggestions. We include the referees and editors' comment below in black font, our responses in blue font, and the changes made in-text (and/or their line numbers in the revised manuscript) in italics blue font.

Best regards,

On behalf of all co-authors,

Sylvain Monteux

**Response to Associate Editor and Co-Editor in Chief decisions**

Dear Authors,

Thank you for replying to all of the comments provided by the referees. Both referees found that your manuscipt is overall very well written, but in addition to some clarifications urge you to focus more on highlighting the novelty of your findings throughout the text. Please proceed to upload a revised version of the manuscript once you have made the required edits.

Best regards,

Emily Solly

In endorse the handling editor's decision and agree that it will be important to highlight the novelty and the relevance of your findings more clearly.

Best regards,

Michael Bahn

We have improved the tone of our manuscript in response to the referees' comments, which we believe addresses the editors' concerns regarding the significance and novelty of our findings.

*More specifically, the changes made to the abstract (lines 21-24), discussion (lines 290-324) and conclusion (lines 381-382) now emphasize better the importance of our findings, and their relevance to soil ecology, biogeochemistry and the permafrost climate feedback.*

The most significant change in our revised manuscript, beyond the abovementioned changes of tone and emphasis, consists in the addition of a conceptual figure depicting our experimental design, to address concerns from the referees about the clarity of our experimental design.

We also now include a missing affiliation for SM and EJK, which was omitted from the initial version of the manuscript.

**Response to referee #1 on comment BG-2022-98-RC1**

General comments

The paper addresses the impact of Collembola presence on bacterial communities in permafrost soils, and its impact on CO2 production, which is a relevant topic for publication in BG. Overall this is a well-written

manuscript, with that is easy to follow, and the Authors do a good job in keeping a focussed approach on the main research hypothesis. I see a few aspects which need to be explained or clarified, and have provided my comments on these as well as some other suggested changes to improve clarity, see below.

We thank the referee for their appreciation of our manuscript and for their comments which helped us improve the text.

Specific comments

ABSTRACT

Line 20-23: this sentence is unclear. What do you mean with different treatments? Do you mean growth substrate / spraying? Also, it is not clear what it means "a number of introduced bacteria were found". Do you mean presence/ absence?

We indeed refer to the distinct growth substrate and spraying manipulation when mentioning treatments, and to the higher number of bacterial ASVs when collembola were present. We have now clarified this section as follows.

*L21-24: The presence of collembola introduced bacterial ASVs absent in the no-collembola control, regardless of their microbiome manipulation, when considering presence-absence metrics (unweighted UniFrac metrics), which resulted in increased species richness. However, these introduced ASVs did not induce changes in bacterial community composition as a whole (accounting for relative abundances, weighted UniFrac), which might only become detectable in the longer-term.*

Line 25: how was estimated that 13.22% of CO2 production was a result of priming effect by collembolan? Shortly explain it here.

We now explain this point in the previous sentence:

*L25-26: $CO_2$ production was increased by 25.85% in the presence of collembola, about half of which could be attributed to collembola respiration based on respiration rates measured in absence of soil.*

Moreover, it is not clear if there is a different effect of collembola on permafrost vs. topsoil, both in terms of CO2 production and bacterial community.

Testing for the effects of collembola on $CO_2$ production or bacterial community on the topsoil, where collembola assemblages are natively present in high abundances, was beyond the scope of our study and such results are therefore not presented. While most studies on the impact of collembola on CO2 production have focused on litter rather than soil, we refer to several studies on soil in the Discussion at lines 346-352.

INTRODUCTION

• Line 36: can you add reference for this sentence?

This sentence builds up on a) the scarcity or absence of macro-invertebrate decomposers such as millipedes, woodlice or earthworms, b) the high abundances of microbivorous fauna in arctic systems (collembola and nematodes), and c) the relation between abundance of microbivorous fauna and microbial decomposition. We understand that this implicit reasoning was not obvious in the current phrasing, and have now clarified and referenced these points as follows.

*L36-39: In arctic soils, the scarcity of macrofaunal decomposers (e.g. earthworms, Blume-Werry et al., 2020) coupled to the high abundance of microbivorous microarthropods such as collembola (Potapov et al., 2022), results in a particularly strong impact of collembola on decomposition through microbial population control (Koltz et al., 2018, Crowther et al., 2012, Seastedt 1984).*

- Line 74: The research gap you are trying to fill should be more clearly addressed, in order to better link this paragraph with the next one (from line 75-80). Here you write "To our best knowledge, this has not been explored yet". However, it is not clear what exactly has not been explored. Could you clarify it here?

The research gap was indeed not clear enough, and has now been rephrased as follows.

*L75-76: To our best knowledge, whether collembola affect the biogeochemical functioning of newly-thawed permafrost, and whether and how they can serve as a vector for microbial colonization has not been explored yet.*

- Line 75: it would be useful to clarify where the Yedoma domain is located

We have now added this information as well as an additional reference synthesizing knowledge on deep carbon stocks in Yedoma permafrost deposits.

*L78-79: We incubated permafrost from the Yedoma domain, which represents a large carbon stock in parts of Siberia and North-America (Strauss et al., 2017) […]*

- Line 81: At point 1, I would specify the collembola presence in permafrost/topsoils

L84: We have now clarified this point according to the referee's suggestion.

RESULTS

- Figure 2: write out in the figure caption the full meaning of the abbreviation ASVs

We now explain the abbreviation ASV throughout the figures.

- Line 264-266: where are the results shown? Add reference to Figure or Table in main text or Appendix

We have now clarified the text so that this sentence and the following both refer to Appendixes A2 and A3.

- Appendix A1: clarify that "RM-ANOVA" means repeated-measures ANOVA

We have clarified this point according to the referee's suggestion, by removing the abbreviation altogether.

DISCUSSION

- Line 340-343: unclear sentence, which could be shortened (e.g. three studies)

We have now streamlined this sentence.

*L346-349: The scarce studies on soil $CO_2$ production including collembola reported contrasted findings, with some studies observing increases in soil $CO_2$ emissions in presence of collembola by up to +400% (Addison and Parkinson, 1978; Bakonyi, 1989; Kaneda and Kaneko, 2008; Wang et al., 2017), while others found no significant changes (Theenhaus et al., 1999; Lubbers et al., 2020; Lucas et al., 2020)*

Technical corrections

We have now improved the following sentences:

• 31: "soil fauna are" to " soil fauna is"

• 32: "this decomposition" to "organic matter decomposition"

• 40: "they" to "collembola"

• 43: consider about shortening the sentence "often considered from the perspective of the huge…."

• 45: "such organisms such as" to "organisms such as"

• 47: consider about shortening the sentence : "Despite this particularity, in the very active field of permafrost research the absence …."

• 65. "In active layer deepening" to " with the deepening of active layer"

• 289: "It may be that …" to e.g. "this could be attributed…"

• 340: "their effect"? Could you check this, it is unclear to what the effect it is referred to.

**Response to referee #1 on comment bg-2022-98-RC2**

Dear Authors,

I have an additional comment on the Manuscript. In the method section, it is not clear how many replicates per treatment (corresponding to the number of jars per treatment) are used. In Figure 2 and 3, it is reported that n = 6, but this is not mentioned in the method section, thus a clarification on this point is needed.

We thank the reviewer for this remark, we indeed omitted to mention this important information in the main text. We now mention the number of replicates in the Methods section when describing the experimental design, and also include it in the new conceptual Figure 1 depicting the experimental design.

**Response to referee #2 on comment BG-2022-98-RC3**

General comments:

The manuscript by Monteux et al is very well written and the described mesocosm experiment is sound and thoughtful. Unfortunately, the results did not match the expectations, but this does not reduce the scientific quality of the manuscript.

We thank the referee for their commendation of our manuscript, and for their comments which helped us improve the text.

The experiment was designed to analyse the effects of presence of collembola on bacteria communities in newly thawed permafrost soil. Permafrost soil was cultured in vessels and collembola (F. candida) were added either from stock cultures on gypsum medium, permafrost medium or an artic soil medium. The collembola were fed with either yeast or the natural community in the soil communities (to explore endozoochory), or were sprayed with suspension from permafrost soil or artic soil (ectozoochory). The presence of Collembola resulted in significant changes in the bacterial community for the ectozoochory and ectozoochory control treatment only, and only when analysed qualitatively (presence/absence), but not quantitatively (community composition). $CO_2$ production was higher by 25% in presence of collembola (of which half could be attributed to a priming effect of Collembola on permafrost soil), but did not vary between Collembola treatments. These results are less impressive than expected and are summarized in a very pointed way in the very first lines of the discussion.

Further in the discussion the authors list a number of reasons why the results were the way they were: bacteria are not able to survive gut passage or on collembola skin, Collembola prefer fungi, etc. This is all true but makes the reader believe that the experiment just failed (sorry, that was my strongest impression).

In my opinion, the results should be stated in a more positive way. Collembola sprayed with arctic soil (and sprayed with permafrost, too) did change the bacterial community, and that is something! Obviously the community composition per se did not change within this short time frame, but the introduction of new microbial species was successful (in future experiments one could try to find out which strains were transferred successfully).Therefore, with enough time for the introduced microorganisms to multiply, community might change.  One should argue that Collembola changed the microbial community despite the fact that they do not really feed on this group of microorganisms and despite the short time investigated. But it might be discussed why this can only be achieved with bacteria sprayed on the skin of Collembola but not those that travelled in the collembolan gut. Secondly, the presence of Collembola increased $CO_2$ release of soil microorganisms by ~15%. For this part of the discussion the same hints are valid as for the community: be more positive. Collembola presence affected soil microorganisms' activity. But what was it that affected microorgnisms? It had nothing to do with introduction of new species, but maybe it was feeding or trampling by the collembola or predigestion of nutrients?

The authors mention all these things already, but the manuscript's sound is too much of an excuse for not-expected results than presentation of the findings. The authors' concerns are right and prove their ability of scientific writing, but they are too much in the focus in the present manuscript.

We understand the referee's concerns: sounding negative or suggesting that the experiment has failed were not messages we intended to convey. We have now rephrased parts of the discussion to make the tone more positive and better emphasize the importance of our findings, in particular that the limited effect on bacterial community composition might be transient due to our short-term incubation period, and that collembola presence in itself has sizeable effects on $CO_2$ production, regardless of initially minor changes to the bacterial community.

Our changes are found in the discussion throughout lines 280-298, 312-315, 321-324, as well as in the abstract at lines 21-24, and conclusion at lines 381-382.

Specific comments:

- In general, the materials and methods (M&M) chapter could be more clear. I had to read it several times to understand which kind of soil was used for which collembola cultures.

We have now rephrased parts of M&M (lines 115-118 & 126-128) and added a conceptual figure (new Figure 1) describing the experimental design to clarify this point.

- Hypothesis II and IV: Do you expect different bacterial strains transferred by ecto- and endozoochory, or the same strains in differing amounts? This is not clear in the hypothesis or M&M section.

One could indeed expect differences between the strains transferred by ecto- and endozoochory: an assemblage of bacteria driven by stochasticity may be transported on collembola cuticle, while endozoochory could disperse a more narrow group of bacteria shaped by deterministic processes due to the constraints of surviving gut transit. This could be supported by evidence on the dispersal of microorganisms by macroorganisms, which has been suggested to follow intricate patterns, including microorganisms relying on surviving gut transit and using fauna as a dispersal vector (Troussellier et al, 2017, Vašutová et al., 2019).

Nevertheless, in the absence of difference between ecto- and endozoochory treatments, and because we did not find sufficient evidence in the literature, we could not discuss this interpretation without being overly speculative. We therefore did not include the corresponding rationale to avoid creating loose ends which we would not address later. We are, however, open to doing so at the editors' discretion.

- How many replicates did you have? I suggest it's five or six from the graphs in the appendix (the grey dots), but you don't write it in M&M. And how many replicates for the "Collembola without soil"-control?

As pointed out in our Author Comment 2 in response to Referee #1 RC2, we have corrected this oversight and now indicate clearly in the Methods section, as well as in the new conceptual Figure 1, that we used 6 replicates for all treatments, including the "Collembola without soil" control.

- Why did you use permafrost soil from Alaska, but actic soil from Sweden? Wouldn't it be better (or more realistic) to use arctic soil from a closer region? Could the distance between the sampling sites affect the ability of the arctic microflora to adapt to the permafrost soil?

Using active layer soil from Alaska would indeed have been preferable, we could not do that due to practical constraints, as we did not have access to such a soil throughout the Covid-19 events.

The choice of Yedoma permafrost from Alaska rather than permafrost from northern Sweden stems from our previous findings showing both a strong control of microbial community composition on Yedoma permafrost functioning and a vulnerability of this permafrost's microbial community to invasions (Monteux et al., 2020).

*We have now rephrased the Methods section to emphasize this point (line 101).*

Regarding concerns about the ability of the Swedish arctic microflora to adapt to this permafrost soils, we observed in a separate incubation experiment (AC Figure 1 in Author Comment 3, unpubl. data) that this permafrost's bacterial community is vulnerable to coalescence with active layer communities both from Alaska (AL1 and AL2) as well as from the Swedish sub-Arctic (AL3). Because these data belong to an upcoming manuscript and showing them would require extensive additions to the M&M section, we would prefer not including them in the main text.

- Were collembola in ectozoochory treatments obtained from stock cultures or from permafrost soil cultures? Please make the section about your treatments more clear.

The collembola in ectozoochory treatments were from the stock cultures on gypsum without permafrost soil, we have now clarified this point both in text (*lines 115-118 & 126-128*) and through the addition of a conceptual figure as the new *Figure 1*.

Small comments/typos:

- Line 264 "than" instead of "that"

We have now corrected this typo.

- Appendix A2, 2nd table: What do you mean with "vs. control"? The no collembola-control or the respective controls?

We have now clarified that this refers to the no-collembola control.

- Appendix A3: In the figure caption you mix up "RRsoil" and "Rrgross".

We have now corrected this mistake.

**References**

Monteux, S., Keuper, F., Fontaine, S., Gavazov, K., Hallin, S., Juhanson, J., Krab, E. J., Revaillot, S., Verbruggen, E., Walz, J., Weedon, J. T., and Dorrepaal, E.: Carbon and nitrogen cycling in Yedoma permafrost controlled by microbial functional limitations, Nature Geoscience, 13, 794–798, https://doi.org/10.1038/s41561-020-00662-4, 2020.

Strauss, J., Schirrmeister, L., Grosse, G., Fortier, D., Hugelius, G., Knoblauch, C., Romanovsky, V., Schädel, C., Schneider von Deimling, T., Schuur, E. A. G., Shmelev, D., Ulrich, M., and Veremeeva, A.: Deep Yedoma permafrost: A synthesis of depositional characteristics and carbon vulnerability, Earth-Science Reviews, 172, 75–86, https://doi.org/10.1016/j.earscirev.2017.07.007, 2017.

Troussellier, M., Escalas, A., Bouvier, T., and Mouillot, D.: Sustaining Rare Marine Microorganisms: Macroorganisms As Repositories and Dispersal Agents of Microbial Diversity, Frontiers in Microbiology, 8, 2017.

Vašutová, M., Mleczko, P., López-García, A., Maček, I., Boros, G., Ševčík, J., Fujii, S., Hackenberger, D., Tuf, I. H., Hornung, E., Páll-Gergely, B., and Kjøller, R.: Taxi drivers: the role of animals in transporting mycorrhizal fungi, Mycorrhiza, 29, 413–434, https://doi.org/10.1007/s00572-019-00906-1, 2019.